# Metabolic pathway inference using multi-label classification with rich pathway features

**Abdur Rahman M. A. Basher** [1], **Ryan J. McLaughlin** [1], **Steven J. Hallam** [1,2,3,4,5] *

**1** Graduate Program in Bioinformatics, University of British Columbia, Genome Sciences Centre, 100-570 West 7th Avenue, Vancouver, British Columbia, Canada, **2** Department of Microbiology & Immunology, University of British Columbia, 2552-2350 Health Sciences Mall, Vancouver, British Columbia, Canada, **3** Genome Science and Technology Program, University of British Columbia, 2329 West Mall, Vancouver, BC, Canada, **4** Life Sciences Institute, University of British Columbia, Vancouver, British Columbia, Canada, **5** ECOSCOPE Training Program, University of British Columbia, Vancouver, British Columbia, Canada

* shallam@mail.ubc.ca

**Data Availability Statement:** All relevant data are within the manuscript and its Supporting Information files.

**Funding:** This work was performed under the auspices of Genome Canada, Genome British

## Abstract

Metabolic inference from genomic sequence information is a necessary step in determining the capacity of cells to make a living in the world at different levels of biological organization. A common method for determining the metabolic potential encoded in genomes is to map conceptually translated open reading frames onto a database containing known product descriptions. Such gene-centric methods are limited in their capacity to predict pathway presence or absence and do not support standardized rule sets for automated and reproducible research. Pathway-centric methods based on defined rule sets or machine learning algorithms provide an adjunct or alternative inference method that supports hypothesis generation and testing of metabolic relationships within and between cells. Here, we present mlLGPR, *m*ulti-*l*abel based on *l*ogistic re*g*ression for *p*athway p*r*ediction, a software package that uses supervised multi-label classification and rich pathway features to infer metabolic networks in organismal and multi-organismal datasets. We evaluated mlLGPR performance using a corpora of 12 experimental datasets manifesting diverse multi-label properties, including manually curated organismal genomes, synthetic microbial communities and low complexity microbial communities. Resulting performance metrics equaled or exceeded previous reports for organismal genomes and identify specific challenges associated with features engineering and training data for community-level metabolic inference.

## Author summary

Predicting the complex series of metabolic interactions e.g. pathways, within and between cells from genomic sequence information is an integral problem in biology linking genotype to phenotype. This is a prerequisite to both understanding fundamental life processes and ultimately engineering these processes for specific biotechnological applications. A pathway prediction problem exists because we have limited knowledge of the reactions and pathways operating in cells even in model organisms like *Esherichia coli* where the

Columbia, the Natural Sciences and Engineering Research Council (NSERC) of Canada, and Compute/Calcul Canada through grants award to S. J.H. The funders had no role in study design, data collection and analysis, decision to publish, or preparation of the manuscript.

**Competing interests:** SJH is a co-founder of Koonkie Inc., a bioinformatics consulting company that designs and provides scalable algorithmic and data analytics solutions in the cloud.

majority of protein functions are determined. To improve pathway prediction outcomes for genomes at different levels of complexity and completion we have developed mlLGPR, _m_ulti-_l_abel based on _l_ogistic re_g_ression for _p_athway p_r_ediction, a scalable open source software package that uses supervised multi-label classification and rich pathway features to infer metabolic networks. We benchmark mlLGPR performance against other inference methods providing a code base and metrics for continued application of machine learning methods to the pathway prediction problem.

## Introduction

Metabolic inference from genomic sequence information is a fundamental problem in biology with far reaching implications for our capacity to perceive, evaluate and engineer cells at the individual, population and community levels of organization [1, 2]. Predicting metabolic interactions can be described in terms of molecular events or reactions coordinated within a series or cycle. The set of reactions within and between cells defines a reactome, while the set of linked reactions defines pathways within and between cells. Reactomes and pathways can be predicted from primary sequence information and refined using mass spectrometry to both validate known and uncover novel pathways.

The development of reliable and flexible rule sets for metabolic inference is a non-trivial step that requires manual curation to add accurate taxonomic or pathway labels [3]. This problem is compounded by the ever increasing abundance of different data structures sourced from organismal genomes, single-cell amplified gemomes (SAGs) and metagenome assembled genomes (MAGs) (Fig 1). Under ideal circumstances, pathways are inferred from a bounded reactome that has been manually curated to reflect detailed biochemical knowledge from a closed reference genome e.g. T1 in the information hierarchy in (Fig 1). While this is possible for a subset of model organisms, it becomes increasingly difficult to realize when dealing with the broader range of organismal diversity found in natural and engineered environments. At the same time, advances in sequencing and mass spectrometry platforms continue to lower the cost of data generation resulting in exponential increases in the volume and complexity of multi-omic information (DNA, RNA, protein and metabolite) available for metabolic inference [4].

Over the past three decades, several trusted sources have emerged to collect and curate reactomes and pathways based on biochemical knowledge including the Kyoto Encyclopedia of Genes and Genomes (KEGG) [5], Reactome [6], and MetaCyc [7]. MetaCyc is a multi-organism member of the BioCyc collection of Pathway/Genome Databases (PGDB) [8] that contains only experimentally validated metabolic pathways across all domains of life (currently over 2766 pathways from 3067 different organisms). Pathway/Genome Databases can be constructed in Pathway Tools, a production-quality software environment developed at SRI that supports metabolic inference based on the MetaCyc database [9]. Navigable and extensively commented pathway descriptions, literature citations, and enzyme properties combined within a PGDB provide a coherent structure for exploring and interpreting pathways in genomes to biomes. Metabolic inference in Pathway Tools is based on the use of a rule-based algorithm called PathoLogic [10] producing organismal PGDBs e.g. EcoCyc [11] stored in repositories e.g. BioCyc [12] that can be refined based on experimental validation. In addition to organismal PDGBs, pathologic can be used to produce microbiome or environmental Pathway/Genome Databases (ePGDBs) representing community level metabolic models e.g. T4 on

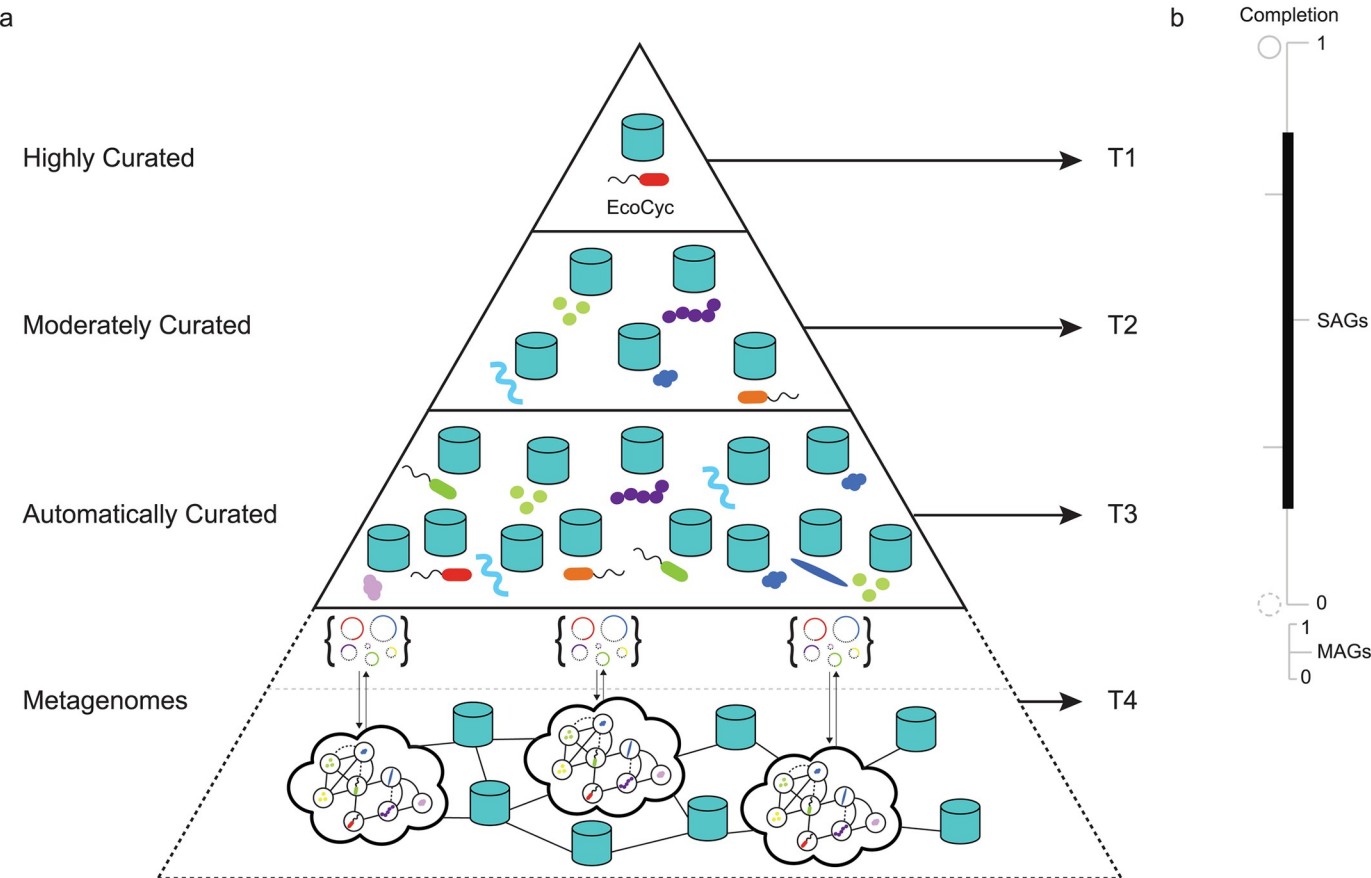

**Fig 1. Genomic information hierarchy encompassing individual, population and community levels of cellular organization.** (a) Building on the BioCyc curation-tiered structure of Pathway/Genome Databases (PGDBs) constructed from organismal genomes, two additional data structures are resolved from single-cell and plurality sequencing methods to define a 4 tiered hierarchy (T1-4) in descending order of manual curation and functional validation. (b) Completion scales for organismal genomes, single-cell amplified gemomes (SAGs) and metagenome assembled genomes (MAGs) within the 4 tiered information hierarchy. Genome completion will have a direct effect on metabolic inference outcomes with incomplete organismal genomes, SAGs or MAGS resolving fewer metabolic interactions.

the information hierarchy in (Fig 1) [13–15] that can also be stored in open source repositories e.g. EngCyc or GutCyc [14, 16].

While PathoLogic provides a powerful engine for pathway-centric inference, it is a hard coded and relatively inflexible application that does not not scale efficiently for community sequencing projects. Moreover, PathoLogic does not provide probability scores associated with inferred pathways further limiting its statistical power with respect to false discovery. An alternative inference method called MinPath uses integer programming to identify the minimum number of pathways that can be described given a set of defined input sequences e.g. KO family annotations in KEGG [17]. However, such a parsimony approach is prone to false negatives and can be difficult to scale. Issues of probability and scale have led to the consideration of machine learning (ML) approaches for pathway prediction based on rich feature information. Dale and colleagues conducted a comprehensive comparison of PathoLogic to different types of supervised ML algorithms including naive Bayes, k nearest neighbors, decision trees and logistic regression, converting PathoLogic rules into features and defining new features for pathway inference [18]. They evaluated these algorithms on experimentally validated pathways from six T1 PGDBs in the BioCyc collection randomly divided into training and test sets.

Resulting performance metrics indicated that generic ML methods equaled or marginally exceeded the performance of PathoLogic with the benefit of probability estimation for pathway presence and increased flexibility and transparency of use.

Despite the potential benefits of adopting ML methods for pathway prediction from genomic sequence information, PathoLogic remains the primary inference engine of Pathway Tools [9], and alternative methods for pathway-centric inference expanding on the algorithms evaluated by Dale and colleagues remain nascent. Several recent efforts incorporate metabolite information to improve pathway inference and reaction rules to infer metabolic pathways [3, 19–21]. Others, including BiomeNet [22] and MetaNetSim [23] omit pathways and model reaction networks based on enzyme abundance information. Here we describe a multi-label classification approach to metabolic pathway inference using rich pathway feature information called mlLGPR, _m_ulti-_l_abel based on _l_ogistic re_g_ression for _p_athway p_r_ediction. mlLGPR uses logistic regression and feature vectors inspired by the work of Dale and colleagues to predict metabolic pathways for individual genomes as well as more complex cellular communities e.g. microbiomes. We evaluate mlLGPR performance in relation to other inference methods including PathoLogic and MinPath on a set of T1 PGDBs alone and in combination from the BioCyc collection, symbiont genomes encoding distributed metabolic pathways for amino acid biosynthesis [24], genomes used in the Critical Assessment of Metagenome Interpretation (CAMI) initiative [25], and whole genome shotgun sequences from the Hawaii Ocean Time Series (HOTS) [26].

## The mlLGPR method

In this section, we provide a series of definitions and the problem formulation followed by a description of mlLGPR components (Fig 2) including: i)- features representation, ii)- the prediction model, and iii)- the multi-label learning process. mlLGPR was written in Python v3 and depends on scikit-learn v0.20 [27], Numpy v1.16 [28], NetworkX v2.3 [29], and SciPy v1.4 [30]. The mlLGPR workflow is presented in (Fig 1).

## Definitions and problem formulation

Here, the default vector is considered to be a column vector and is represented by a boldface lowercase letter (e.g., $\mathbf{x}$) while the matrix of it is denoted by boldface uppercase letter (e.g., $\mathbf{X}$). Unless otherwise mentioned, if a subscript letter $i$ is attached to a matrix, such as $\mathbf{X}_i$, it indicates the $i$-th row of $\mathbf{X}$, which is a row vector while a subscript character to a vector, $\mathbf{x}_i$, represents an $i$-th cell of $\mathbf{x}$. Occasional superscript, $\mathbf{x}^{(i)}$, suggests an index to a sample or current epoch during learning period. With these notations in mind, we introduce the metabolic pathway inference problem by first defining the pathway dataset.

Metabolic pathway inference can be formulated as a supervised multi-label prediction problem. This is because a genome encodes multiple pathway labels per instance. Formally, let $\mathcal{S} = \{(\mathbf{x}^{(i)}, \mathbf{y}^{(i)}) : 1 < i \leqslant n\}$ be a pathway dataset consisting of $n$ examples, where $\mathbf{x}^{(i)}$ is a vector indicating abundance information for corresponding enzymatic reactions. An enzymatic reaction is denoted by $e$, which is an element of a set $\mathcal{E} = \{e_1, e_2, \ldots, e_r\}$, having $r$ possible enzymatic reactions, hence, the vector size $\mathbf{x}^{(i)}$ is $r$. The abundance of an enzymatic reaction for an example $i$, say $e_l^{(i)}$, is defined as $a_l^{(i)} (\in \mathbb{R}_{\geq 0})$. The class labels $\mathbf{y}^{(i)} = [y_1^{(i)}, \ldots, y_t^{(i)}] \in \{0, 1\}^t$ is a pathway label vector of size $t$ that represents the total number of pathways, which are derived from a set of universal metabolic pathway $\mathcal{Y}$. The matrix form of $\mathbf{x}^{(i)}$ and $\mathbf{y}^{(i)}$ are $\mathbf{X}$ and $\mathbf{Y}$, respectively.

We further denote $\mathcal{X} = \mathbb{R}^d$ as the $d$-dimensional input space, and transform each sample $\mathbf{x}^{(i)} \in \mathcal{X}$ into an arbitrary $m$-dimensional vector based on a transformation function where

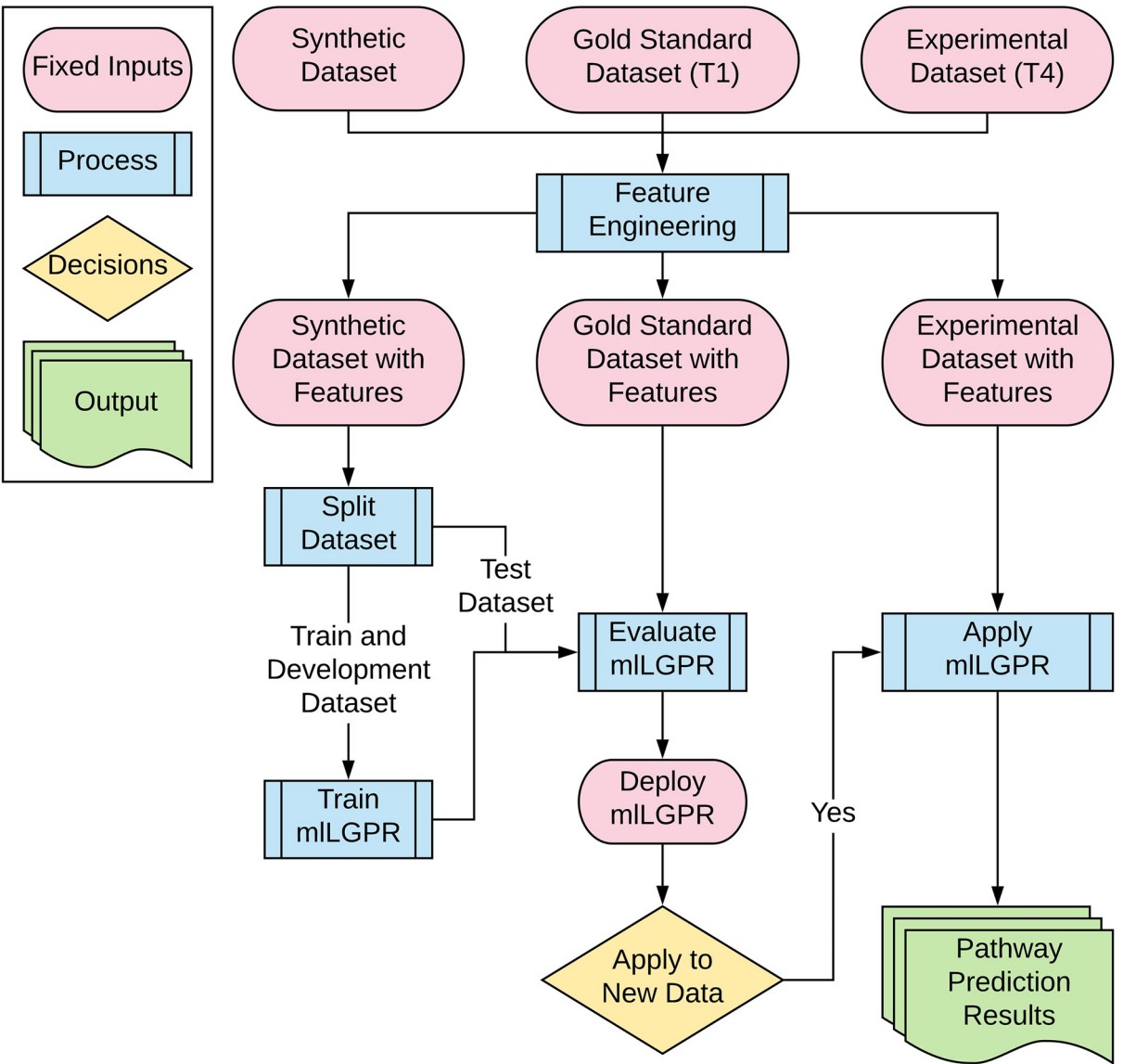

**Fig 2. mlLGPR workflow.** Datasets spanning the information hierarchy are used in feature engineering. The Synthetic dataset with features is split into training and test sets and used to train mlLGPR. Test data from the Gold Standard dataset (T1) with features and Synthetic dataset (T1-3) with features is used to evaluate mlLGPR performance prior to the application of mlLGPR on experimental datasets (T4) from different sources.

$m \gg d$. The transformation function for each sample $i$ is defined by $\Phi : \mathcal{X} \rightarrow \mathbb{R}^m$, which can be described as a feature extraction and transformation process (see Section Features engineering). Given the above notation and a multi-label dataset $\mathcal{S}$, we want to learn a hypothesis function $f : \Phi(\mathbf{x}) \rightarrow 2^{|\mathcal{Y}|}$ from $\mathcal{S}$, such that it predicts metabolic pathways in new samples as accurately as possible.

## Features engineering

The design of feature vectors is critical for accurate classification and pathway inference. We consider five types of feature vectors inspired by the work of Dale and colleagues [18]: i)-

enzymatic reactions abundance vector ($\phi^a$), ii- reactions evidence vector ($\phi^f$), iii- pathways evidence vector ($\phi^y$), iv- pathway common vector ($\phi^c$), and v- possible pathways vector ($\phi^d$). The transformation process $\phi^a$ is represented by $r$-dimensional frequency vector, corresponding to the number of occurrences for each enzymatic reaction as $\phi^a = [a_1, a_2, \ldots, a_r]^\top$. An enzymatic reaction is characterized by an enzyme commission (EC) classification number [31]. The reaction evidence vector $\phi^f$ indicates the properties of the enzymatic reaction for each sample. The pathway evidence features $\phi^y$ include a modified subset of features developed by Dale and colleagues expanding on core PathoLogic rule sets to include additional information related to enzyme presence, gaps in pathways, network connectivity, taxonomic range, etc [18]. The pathway common feature vector $\phi^c$, for a sample $\mathbf{x}^{(i)}$ is represented by $r$-dimensional binary vector and the possible pathways vector $\phi^d$ is a $t$-dimensional binary vector. Each of the transformation function maps $\mathbf{x}$ to a different dimensional vector, and the concatenated feature vector $\Phi = [\phi^a(\mathbf{x}^{(i)}), \phi^f(\mathbf{x}^{(i)}), \phi^y(\mathbf{x}^{(i)}), \phi^c(\mathbf{x}^{(i)}), \phi^d(\mathbf{x}^{(i)})]$ has a total of $m$-dimensional features for each sample. For a more in-depth description of the feature engineering process please refer to S2 Appendix).

## Prediction model

We use the logistic regression (LR) model to infer a set of pathways given an instance feature vector $\Phi(\mathbf{x}^{(i)})$. LR was selected because of its proven power in discriminative classification across a variety of supervised machine learning problems [32]. In addition to direct probabilistic interpretation integrated into the model, LR can handle high-dimensional data, efficiently. The LR model represents conditional probabilities through a non-linear logistic function $f(.)$ defined as

$$f(\theta_j, \Phi(\mathbf{x}^{(i)})) = p(\mathbf{y}_j^{(i)} = 1 | \Phi(\mathbf{x}^{(i)}); \theta_j) = \frac{\exp(\theta_j^\top \Phi(\mathbf{x}^{(i)}))}{\exp(\theta_j^\top \Phi(\mathbf{x}^{(i)})) + 1} \tag{1}$$

where $\mathbf{y}_j^{(i)}$ is the $j$-th element of the label vector $\mathbf{y}^{(i)} \in \{0, 1\}^t$ and $\theta_j$ is a $m$-dimensional weight vector for the $j$-th pathway. Each element of $\Phi(\mathbf{x}^{(i)})$ corresponds to an element of $\theta_j$ for the $j$-class, therefore, we can retrieve important features that contribute to the prediction of $j$ by sorting the elements of $\Phi(\mathbf{x}^{(i)})$ according to the corresponding values of the weight vector $\theta_j$. The Eq 1 is repeated for all the $t$ classes for an instance $i$, hence multi-labeling, and, for an individual pathway, the results are stored in a vector $\mathbf{q}^{(i)} \in \mathbb{R}^t$. Predicted pathways are reported based on a cut-off threshold $\tau$, which is set to 0.5 by default:

$$\widehat{\mathbf{y}^{(i)}} = \text{vec}\left( \begin{cases} 1 & \text{if } \mathbf{q}_j^{(i)} \geq \tau \\ 0 & \text{otherwise} \end{cases} \right) \forall j \in t \tag{2}$$

where vec is a vectorized operation. Given that Eq 1 produces a conditional probability over each pathway, and the $j$-th class label will be included to $\mathbf{y}^{(i)}$ only if $f(\theta_j, \Phi(\mathbf{x}^{(i)})) \geq \tau$ we adopt a soft decision boundary using T-criterion rule [33] as:

$$\widehat{\mathbf{y}^{(i)}} = \text{vec}\left( \begin{cases} 1 & \text{if } \mathbf{q}_j^{(i)} \geq \tau \\ 1 & \text{if } \mathbf{q}_j^{(i)} \geqslant f_{max}(\mathbf{q}_j^{(i)}) \\ 0 & \text{otherwise} \end{cases} \right) \forall j \in t \tag{3}$$

where $f_{max}(f(\theta_j, \Phi(\mathbf{x}^{(i)}))) = \beta \cdot \max(\{f(\theta_j, \Phi(\mathbf{x}^{(i)}): \forall j \in t\})$, which is the maximum predictive probability score. The hyper-parameter $\beta \in (0, 1]$ must be tuned based on empirical

information, and it cannot be set to 0, which implies retrieving all of the $t$ pathways. The predicted set of pathways using the Eq 3 is referred to as adaptive prediction because the decision boundary, and its corresponding threshold, are tuned to the test data [34].

## Multi-label learning process

The process is decomposed into $t$ independent binary classification problems, where each binary classification problem corresponds to a possible pathway in the label space. Then, LR is used to define a binary classifier $f(.)$, such that for a training example $(\Phi(\mathbf{x}^{(i)}), \mathbf{y}^{(i)})$, an instance $\Phi(\mathbf{x}^{(i)})$ will be involved in the learning process of $t$ binary classifiers. Given $n$ training samples, we attempt to estimate all the weight vectors individually $\theta_1, \theta_2, \ldots, \theta_t$ by maximizing the logistic loss function as follows:

$$ll(\theta_j) = \max_{\theta_j} \frac{1}{n} \sum_{i=1}^{n} (\mathbf{y}_j^{(i)} \theta_j^{\top} \Phi(\mathbf{x}^{(i)}) - \log(1 + \exp(\theta_j^{\top} \Phi(\mathbf{x}^{(i)})))) \tag{4}$$

Usually, a penalty or regularization term $\Omega(\theta_j)$ is inserted into the loss function to enhance the generalization properties to unseen data, particularly if the dimension $m$ of features is high. Thus, the overall objective cost function (after dropping the maximized term for brevity) is defined as:

$$C(\theta_j) = ll(\theta_j) - \lambda \Omega(\theta_j) \tag{5}$$

where $\lambda > 0$ is a hyper-parameter that controls the trade-off between $ll(\theta_j)$ and $\Omega(\theta_j)$. Here, the regularization term $\Omega(\theta_j)$ is chosen to be the elastic net:

$$\Omega(\theta_j) = \frac{1-\alpha}{2} \|\theta_j\|_2^2 + \alpha \|\theta_j\|_1 \tag{6}$$

The elastic net penalty of Eq 6 is a compromise between the $L_1$ penalty of LASSO (by setting $\alpha = 1$) and the $L_2$ penalty of ridge-regression (by setting $\alpha = 0$) [35]. While the $L_1$ term of the elastic net aims to remove irrelevant variables by forcing some coefficients of $\theta_j$ to 0, leading to a sparse vector of $\theta_j$, the $L_2$ penalty ensures that highly correlated variables have similar regression coefficients. Substituting Eq 6 into Eq 5, yields the following objective function:

$$C(\theta_j) = ll(\theta_j) - \lambda(\frac{1-\alpha}{2} \|\theta_j\|_2^2 + \alpha \|\theta_j\|_1) \tag{7}$$

During learning, the aim is to estimate parameters $\theta_j$ so as to maximize $C(\theta_j)$, which is convex; however, the last term of Eq 7 is non-differentiable, making the equation non-smooth. For the rightmost term, we apply the sub-gradient [36] method allowing the optimization problem to be solved using mini-batch gradient descent (GD) [37]. We initialize with random values for $\theta_j$, followed by iterations to maximize the cost function $C(\theta_j)$ with the following derivatives:

$$\frac{\partial}{\partial \theta_j} C(\theta_j) = \frac{1}{n} \sum_{i=1}^{n} \Phi(\mathbf{x}^{(i)})[\mathbf{y}_j^{(i)} - f(\theta_j, \Phi(\mathbf{x}^{(i)}))] - \lambda[(1-\alpha)\theta_j + \alpha \ \text{sign}(\theta_j)] \tag{8}$$

Finally, the update algorithm for $\theta_j$ at each iteration is obtained as:

$$\theta_j^{u+1} = \theta_j^{u} + \eta(\frac{1}{n} \sum_{i=1}^{n} \Phi(\mathbf{x}^{(i)})[\mathbf{y}_j^{(i)} - f(\theta_j, \Phi(\mathbf{x}^{(i)}))] - \lambda[(1-\alpha)\theta_j + \alpha \ \text{sign}(\theta_j)]) \tag{9}$$

where $u$ is the current step. The mathematical derivation of the algorithm can be found in S1 Appendix.

## Experimental setup

In this section, we describe an experimental framework used to demonstrate mlLGPR pathway prediction performance across multiple datasets spanning the genomic information hierarchy (Fig 1). MetaCyc version 21 containing 2526 base pathways and 3650 enzymatic reactions, was used as a trusted source to generate samples, build features, and validate results from the prediction algorithms, as outlined in Section Results. For training we constructed two synthetic datasets *Synset* 1 and *Synset* 2 based on the Poisson distribution to subsample pathways, aligning with the previous work [22], from a list of MetaCyc pathways.

We evaluated mlLGPR performance using a corpora of 12 experimental datasets manifesting diverse multi-label properties, including manually curated organismal genomes, synthetic microbial communities and low complexity microbial communities. The T1 golden dataset consisted of six PGDBs including *AraCyc*, *EcoCyc*, *HumanCyc*, *LeishCyc*, *TrypanoCyc*, and *YeastCyc*, A composite golden dataset, referred to as *SixDB*, consisted of 63 permuted combinations of T1 PGDBs. In addition to datasets derived from the BioCyc collection, we evaluated performance using low complexity data from *Moranella* (GenBank NC-015735) and *Tremblaya* (GenBank NC-015736) symbiont genomes encoding distributed metabolic pathways for amino acid biosynthesis [24], the Critical Assessment of Metagenome Interpretation (CAMI) initiative low complexity dataset [25], and whole genome shotgun sequences from the Hawaii Ocean Time Series (HOTS) at 25m, 75m, 110m (sunlit) and 500m (dark) ocean depth intervals [26]. More information about the datasets are summarized in S3 Appendix.

mlLGPR performance was compared to four additional prediction methods including BASELINE, Naïve v1.2 [17], MinPath v1.2 [17] and PathoLogic v21 [10]. In the BASELINE method, the enzymatic reactions of an example $\mathbf{x}^{(i)}$ are mapped directly onto the true representation of all known pathways $\mathcal{Y}$. Then, we apply a cutoff threshold (0.5) to retrieve a list of pathways for that example. In the Naïve method, reactions are randomly predicted from MetaCyc and linked together to construct pathways that are accepted or rejected based on a specified cut-off threshold, typically set to 0.5. If one or more enzymatic reactions are assigned to a pathway then that pathway is identified as present; otherwise, it is rejected. MinPath recovers the minimal set of pathways that can explain observed enzymatic reactions through an iterative constrained optimization process using an integer programming algorithm [38]. PathoLogic is a rule-based metabolic inference method incorporating manually curated biochemical information in a two step process that first produces a reactome that is in turn used to predict metabolic pathways within a PGDB [10].

For training purposes *Synset-1* and *Synset-2*, were subdivided in three subsets: (*training set*, *validation set*, and *test set*), using a stratified sampling approach [39] resulting in 10, 869 training, 1938 validation and 2193 testing samples for Synset-1 and 10, 813 training, 1, 930 validation, and 2, 257 instances for Synset-2. Features extraction was implemented for each dataset in Table 1, resulting in total feature vector size of 12452 for each instance, where $|\phi^a| = 3650$, $|\phi^f| = 68$, $|\phi^y| = 32$, $|\phi^c| = 3650$, and $|\phi^d| = 5052$. Integral hyper-parameter settings included $\Theta$ initialized to a uniform random value in the range [0, 1], batch-size set to 500, epoch number set to 3, adaptive prediction hyper-parameter $\beta$ in the range (0, 1], regularization hyper-parameters $\lambda$ and $\alpha$ set to 10000 and 0.65, respectively. The learning rate $\eta$ was adjusted based on $\frac{1}{\lambda+u}$, where $u$ denotes the current step. The development set was used to determine critical values of $\lambda$ and $\alpha$. Default parameter settings were used for MinPath and PathoLogic. All tests were conducted using a Linux server using 10 cores on an Intel Xeon CPU E5-2650.

**Table 1. Experimental dataset properties.** The notations $|\mathcal{S}|$, $L(\mathcal{S})$, $LCard(\mathcal{S})$, $LDen(\mathcal{S})$, $DL(\mathcal{S})$, and $PDL(\mathcal{S})$ represent number of instances, number of pathway labels, pathway labels cardinality, pathway labels density, distinct pathway labels set, and proportion of distinct pathway labels set for $\mathcal{S}$, respectively. The notations $R(\mathcal{S})$, $RCard(\mathcal{S})$, $RDen(\mathcal{S})$, $DR(\mathcal{S})$, and $PDR(\mathcal{S})$ have similar meanings as before but for the enzymatic reactions $\mathcal{E}$ in $\mathcal{S}$. $PLR(\mathcal{S})$ represents a ratio of $L(\mathcal{S})$ to $R(\mathcal{S})$. The last column denotes the domain of $\mathcal{S}$.

| Dataset | $\|\mathcal{S}\|$ | $L(\mathcal{S})$ | $LCard$ $(\mathcal{S})$ | $LDen$ $(\mathcal{S})$ | $DL$ $(\mathcal{S})$ | $PDL(\mathcal{S})$ | $R(\mathcal{S})$ | $RCard(\mathcal{S})$ | $RDen$ $(\mathcal{S})$ | $DR$ $(\mathcal{S})$ | $PDR(\mathcal{S})$ | $PLR$ $(\mathcal{S})$ | Domain |
|---|---|---|---|---|---|---|---|---|---|---|---|---|---|
| EcoCyc | 1 | 307 | 307 | 1 | 307 | 307 | 1134 | 1134 | 1 | 719 | 719 | 0.2707 | Escherichia coli K-12 substr. MG1655 |
| HumanCyc | 1 | 279 | 279 | 1 | 279 | 279 | 1177 | 1177 | 1 | 693 | 693 | 0.2370 | Homo sapiens |
| AraCyc | 1 | 510 | 510 | 1 | 510 | 510 | 2182 | 2182 | 1 | 1034 | 1034 | 0.2337 | Arabidopsis thaliana |
| YeastCyc | 1 | 229 | 229 | 1 | 229 | 229 | 966 | 966 | 1 | 544 | 544 | 0.2371 | Saccharomyces cerevisiae |
| LeishCyc | 1 | 87 | 87 | 1 | 87 | 87 | 363 | 363 | 1 | 292 | 292 | 0.2397 | Leishmania major Friedlin |
| TrypanoCyc | 1 | 175 | 175 | 1 | 175 | 175 | 743 | 743 | 1 | 512 | 512 | 0.2355 | Trypanosoma brucei |
| SixDB | 63 | 37295 | 591.9841 | 0.0159 | 944 | 14.9841 | 210080 | 3334.6032 | 0.0159 | 1709 | 27.1270 | 0.1775 | Composed from six databases |
| Symbiont | 3 | 119 | 39.6667 | 0.3333 | 59 | 19.6667 | 304 | 101.3333 | 0.3333 | 130 | 43.3333 | 0.3914 | Composed of Moranella and Tremblaya |
| CAMI | 40 | 6261 | 156.5250 | 0.0250 | 674 | 16.8500 | 14269 | 356.7250 | 0.0250 | 1083 | 27.0750 | 0.4388 | Simulated microbiomes of low complexity |
| HOT | 4 | 2178 | 311.1429 | 0.1429 | 781 | 111.5714 | 182675 | 26096.4286 | 0.1429 | 1442 | 206.0000 | 0.0119 | Metagenomic Hawaii Ocean Time-series (10m, 75m, 110m, and 500m) |
| Synset-1 | 15000 | 6801364 | 453.4243 | 0.00007 | 2526 | 0.1684 | 30901554 | 2060.1036 | 0.00007 | 3650 | 0.2433 | 0.2201 | Synthetically generated (uncorrupted) |
| Synset-2 | 15000 | 6806262 | 453.7508 | 0.00007 | 2526 | 0.1684 | 34006386 | 2267.0924 | 0.00007 | 3650 | 0.2433 | 0.2001 | Synthetically generated (corrupted) |

## Performance metrics

The following metrics were used to report on performance of prediction algorithms used in the experimental framework outlined above: *average precision*, *average recall*, *average F1 score (F1)*, and *Hamming loss*, [40].

Formally, let us denote $\mathbf{y}^{(i)}$ and $\widehat{\mathbf{y}^{(i)}}$ to be the true and the predicted pathway set for the *i*-the sample, respectively. Then, the four measurements can be defined as:

$$\text{Average Precision (Pr)} = \frac{1}{n}\sum_{i=1}^{n}\left(\frac{\mathbf{y}^{(i)\top}\widehat{\mathbf{y}^{(i)}}}{\sum_{j\in t}\widehat{\mathbf{y}}_{j}^{(i)}}\right) \tag{10}$$

$$\text{Average Recall (Rc)} = \frac{1}{n}\sum_{i=1}^{n}\left(\frac{\mathbf{y}^{(i)\top}\widehat{\mathbf{y}^{(i)}}}{\sum_{j\in t}\mathbf{y}_{j}^{(i)}}\right) \tag{11}$$

$$\text{Average F1} = \frac{2\text{Pr}\times\text{Rc}}{\text{Pr}+\text{Rc}} \tag{12}$$

$$\text{Hamming Loss (hloss)} = \frac{1}{nt}\sum_{i=1}^{n}\sum_{j=1}^{t}1(\mathbf{y}_{j}^{(i)}\neq\widehat{\mathbf{y}}_{j}^{(i)}) \tag{13}$$

where 1(.) denotes the indicator function, respectively. Each metric is averaged based on sample size.

The values of *average precision*, *average recall*, and *average F1* vary between $0 - 1$ with 1 being the optimal score. Average Precision relates the number of true pathways to the number

of predicted pathways including false positives, while recall relates the number of true pathways to the total number of expected pathways including false negatives. While recall tells us about the ability of each prediction method to find relevant pathways, precision tells us about the accuracy of those predictions. Average F1 represents the harmonic mean of average precision and average recall by taking the trade-off between the two metrics into account. The hloss is the fraction of pathways that are incorrectly predicted providing a useful performance indicator. From Eq 13, we observe that when all of the pathways are correctly predicted, then hloss = 0, whereas the other metrics will be equal to 1. On the other hand, when the predictions of all pathways are completely incorrect hloss = 1, whereas the other metrics will be equal to 0.

## Results

Four types of analysis including parameter sensitivity, features selection, robustness, and pathway prediction potential were used to tune and evaluate mlLGPR performance in relation to other pathway prediction methods.

### Parameter sensitivity

**Experimental setup.**   Three consecutive tests were performed to ascertain: 1)- the impact of L1, L2, and elastic-net (EN) regularizers on mlLGPR performance using T1 golden datasets, 2)- the impact of changing hyper-parameter $\lambda \in \{1, 10, 100, 1000, 10000\}$ using T1 golden datasets, and 3)- the impact of adaptive beta $\beta \in (0, 1]$ using Synset-2 and the SixDB golden dataset.

**Experimental results.**   Table 2 indicates test results across different mlLGPR configurations. Although the F1 scores of mlLGPR-L1, mlLGPR-L2 and mlLGPR-EN were comparable, precision and recall scores were inconsistent across the T1 golden datasets. For example,

**Table 2. Predictive performance of mlLGPR on T1 golden datasets.** mlLGPR-L1: the mlLGPR with L1 regularizer, mlLGPR-L2: the mlLGPR with L2 regularizer, mlLGPR-EN: the mlLGPR with elastic net penalty, L2: AB: abundance features, RE: reaction evidence features, and PE: pathway evidence features. For each performance metric, '↓' indicates the lower score is better while '↑' indicates the higher score is better.

| Methods | Hamming Loss ↓ | | | | | | |
|---|---|---|---|---|---|---|---|
| | EcoCyc | HumanCyc | AraCyc | YeastCyc | LeishCyc | TrypanoCyc | SixDB |
| mlLGPR-L1 (+AB+RE+PE) | 0.0776 | 0.0645 | 0.1069 | 0.0487 | 0.0412 | 0.0602 | 0.1365 |
| mlLGPR-L2 (+AB+RE+PE) | **0.0606** | **0.0515** | 0.1112 | **0.0412** | **0.0234** | **0.0344** | 0.1426 |
| mlLGPR-EN (+AB+RE+PE) | 0.0804 | 0.0633 | **0.1069** | 0.0550 | 0.0380 | 0.0590 | **0.1281** |
| Methods | Average Precision Score ↑ | | | | | | |
| | EcoCyc | HumanCyc | AraCyc | YeastCyc | LeishCyc | TrypanoCyc | SixDB |
| mlLGPR-L1 (+AB+RE+PE) | 0.6253 | 0.6686 | 0.7390 | 0.6815 | 0.4525 | 0.5395 | 0.7391 |
| mlLGPR-L2 (+AB+RE+PE) | **0.7437** | **0.7945** | **0.8418** | **0.7934** | **0.6186** | **0.7268** | **0.8488** |
| mlLGPR-EN (+AB+RE+PE) | 0.6187 | 0.6686 | 0.7372 | 0.6480 | 0.4731 | 0.5455 | 0.7561 |
| Methods | Average Recall Score ↑ | | | | | | |
| | EcoCyc | HumanCyc | AraCyc | YeastCyc | LeishCyc | TrypanoCyc | SixDB |
| mlLGPR-L1 (+AB+RE+PE) | **0.9023** | 0.8244 | 0.7275 | **0.8690** | **0.9310** | **0.8971** | 0.6738 |
| mlLGPR-L2 (+AB+RE+PE) | 0.7655 | 0.7204 | 0.5529 | 0.7380 | 0.8391 | 0.8057 | 0.5211 |
| mlLGPR-EN (+AB+RE+PE) | 0.8827 | **0.8459** | **0.7314** | 0.8603 | 0.9080 | 0.8914 | **0.6904** |
| Methods | Average F1 Score ↑ | | | | | | |
| | EcoCyc | HumanCyc | AraCyc | YeastCyc | LeishCyc | TrypanoCyc | SixDB |
| mlLGPR-L1 (+AB+RE+PE) | 0.7387 | 0.7384 | 0.7332 | 0.7639 | 0.6090 | 0.6738 | 0.6919 |
| mlLGPR-L2 (+AB+RE+PE) | **0.7544** | **0.7556** | 0.6675 | **0.7647** | **0.7122** | **0.7642** | 0.6306 |
| mlLGPR-EN (+AB+RE+PE) | 0.7275 | 0.7468 | **0.7343** | 0.7392 | 0.6220 | 0.6768 | **0.7098** |

high precision scores were observed for mlLGPR-L2 on AraCyc (0.8418) and YeastCyc (0.7934) with low recall scores of 0.5529 and 0.7380, respectively. In contrast, high recall scores were observed for mlLGPR-L1 on AraCyc (0.7275) and YeastCyc (0.8690) with low precision scores of 0.7390 and 0.6815, respectively. The increased recall with reduced precision scores by mlLGPR-L1 indicates a low variance model that may eliminate many relevant coefficients. The impact is especially observed for datasets encoding a small number of pathways as is the case for LeishCyc (87 pathways) and TrypanoCyc (175 pathways). Similarly, the increased precision with reduced recall scores by mlLGPR-L2 is a consequence of the existence of highly correlated features present in the test datasets [41], resulting in a high variance model. The impact is especially observed for LeishCyc and TrypanoCyc suggesting that mlLGPR-L2 performance declines with increasing pathway number. mlLGPR-EN tended to even out the scores relative to mlLGPR-L1 and mlLGPR-L2 providing more balanced performance outcomes.

Based on these results, hyper-parameters $\lambda$ and $\beta$ were tested to tune mlLGPR-EN performance. Fig 3 indicates that the relationship between F1 score and the regularization hyper-parameter $\lambda$ increases monotonically for the T1 golden datasets peaking at $\lambda = 10000$ (having an F1 score of $> 0.6$ for all datasets). For the adaptive $\beta$ test, Fig 4 shows the performance of mlLGPR-EN on Synset-2 test samples across a range of $\beta \in (0, 1]$ values, indicating that this hyper-parameter has minimal impact on performance.

Taken together, parameter testing results indicated that mlLGPR-EN provided the most balanced implementation of mlLGPR, and the regularization hyper-parameter $\lambda$ at 10000 resulted in the best performance for T1 golden datasets. This hyper-parameter should be tuned when applied to new datasets to reduce false positive pathway discovery. Minimal effects on prediction performance were observed when testing the adaptive hyper-parameter $\beta$.

## Features selection

**Experimental setup.**   A series of feature set "ablation" tests were conducted using Synset-2 as a training set in a reverse manner, starting with only reaction abundance features (AB), a

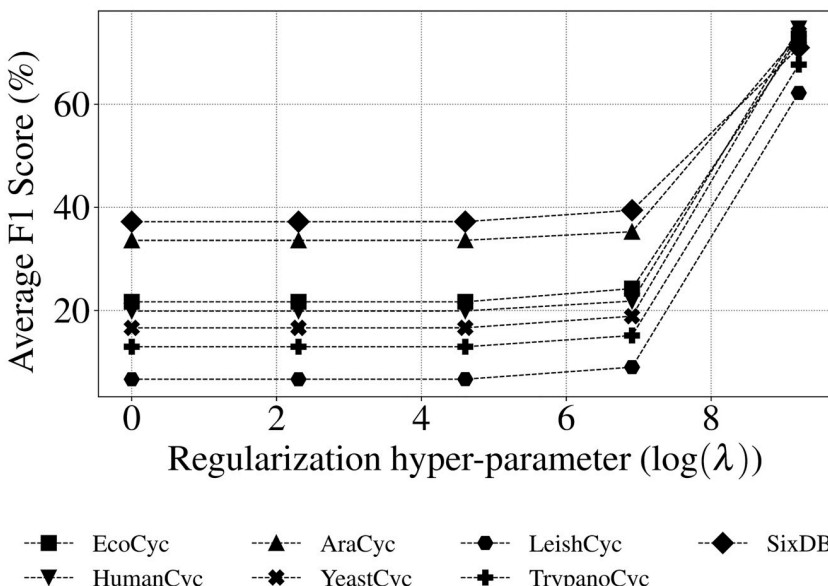

**Fig 3. Average F1 score of mlLGPR-EN on a range of regularization hyper-parameter $\lambda \in \{1, 10, 100, 1000, 10000\}$ values on EcoCyc, HumanCyc, AraCyc, YeastCyc, LeishCyc, TrypanoCyc, and SixDB dataset.** The x-axis is log scaled.

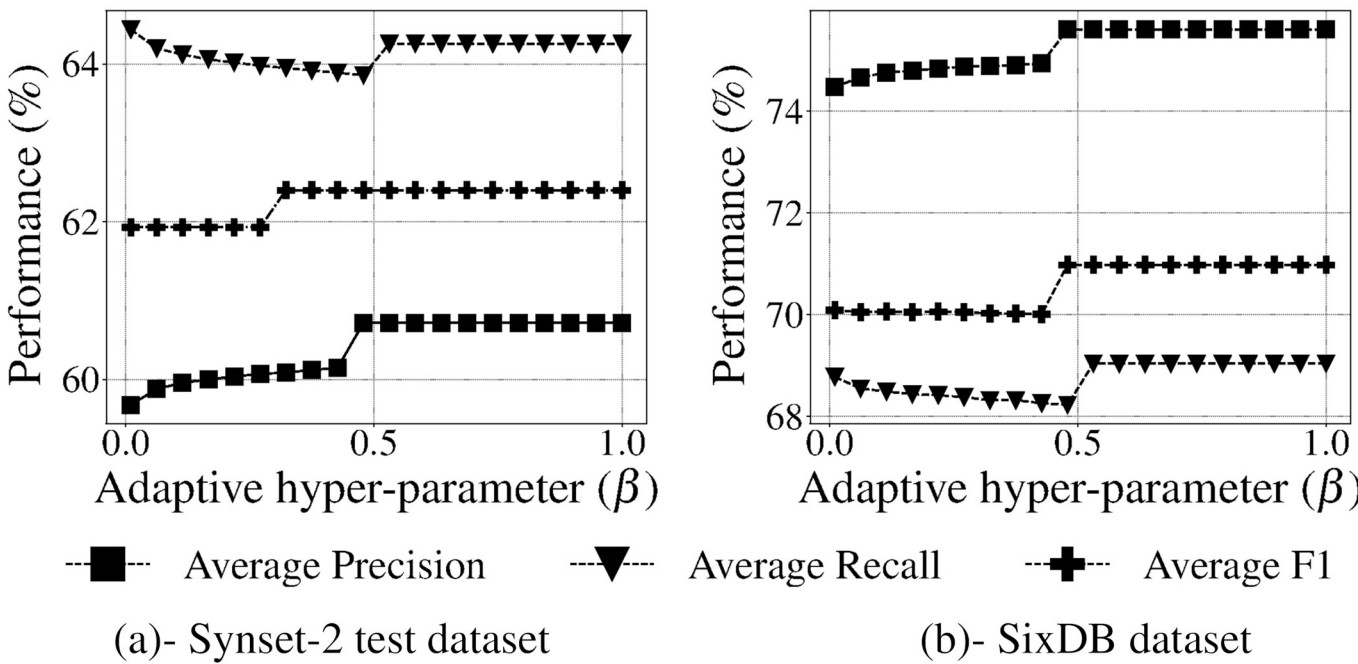

**Fig 4. Performance of mlLGPR-EN according to the $\beta$ adaptive decision hyper-parameter on datasets.** (a)- Synset-2 test dataset. (b)- SixDB dataset.

fundamental feature set consisting of 3650 features and then successively aggregating additional feature sets while recording predictive performance on golden T1 datasets using the settings and metrics described in Section Experimental setup. Because testing individual features is not practical, this form of aggregate testing provides a tractable method to identify the relative contribution of feature sets to pathway prediction performance.

**Experimental results.** Table 3 indicates ablation test results. The AB feature set promotes the highest average recall on EcoCyc (0.9511) and a comparable F1-score of 0.6952. This is not unexpected given the ratio of pathways to the number of enzymatic reactions (PLR) indicated by EC numbers for EcoCyc is high. However, although functional annotations with EC numbers increase the probability of predicting a given pathway, pathways with few or no EC numbers such as *pregnenolone biosynthesis* require additional feature sets to avoid false negatives. As additional feature sets are aggregated, mlLGPR-EN performance tends to improve unevenly for different T1 organismal genomes. For example, adding the enzymatic reaction evidence (RE) feature set consisting of 68 features to the AB features set improves F1 scores for YeastCyc (0.7394), LeishCyc (0.5830), and TrypanoCyc (0.6753). Further aggregating the pathway evidence (PE) feature set, consisting of 32 features to the AB feature set improves the F1 score for AraCyc (0.7532) but reduces the F1 score for the remaining T1 organismal genomes. Aggregating AB, RE and pathway evidence (PE) feature sets resulted in the highest F1 scores for HumanCyc (0.7468), LeishCyc (0.6220), TrypanoCyc (0.6768), and SixDB (0.7078) with only marginal differences between the highest F1 scores for EcoCyc (0.7275) and AraCyc (0.7343). Additional combinations of features did not improve overall performance across the T1 golden datasets.

Taken together, ablation testing results indicated that mlLGPR-EN in combination with AB, RE and PE feature sets result in the most even pathway prediction performance for golden T1 datasets.

**Table 3. Ablation tests of mlLGPR-EN trained using Synset-2 on T1 golden datasets.** AB: abundance features, RE: reaction evidence features, PP: possible pathway features, PE: pathway evidence features, and PC: pathway common features. mlLGPR is trained using a combination of features, represented by mlLGPR-*, on Synset-2 training set. For each performance metric, '↓' indicates the lower score is better while '↑' indicates the higher score is better.

| Methods | Hamming Loss ↓ | | | | | | |
|---|---|---|---|---|---|---|---|
| | EcoCyc | HumanCyc | AraCyc | YeastCyc | LeishCyc | TrypanoCyc | SixDB |
| mlLGPR+AB | 0.1013 | 0.0887 | 0.1025 | 0.0907 | 0.1124 | 0.1073 | 0.1412 |
| mlLGPR+AB+RE | **0.0788** | 0.0697 | 0.1101 | **0.0558** | 0.0447 | **0.0598** | 0.1348 |
| mlLGPR+AB+PP | 0.2835 | 0.2922 | 0.2898 | 0.2724 | 0.2553 | 0.2759 | 0.2842 |
| mlLGPR+AB+PE | 0.1017 | 0.0835 | **0.1002** | 0.0891 | 0.1172 | 0.1089 | 0.1387 |
| mlLGPR+AB+PC | 0.1041 | 0.0938 | 0.1409 | 0.0879 | 0.1081 | 0.0899 | 0.1844 |
| mlLGPR+AB+RE+PP | 0.2815 | 0.2882 | 0.2961 | 0.2648 | 0.2526 | 0.2759 | 0.2825 |
| mlLGPR+AB+RE+PE | 0.0804 | **0.0633** | 0.1069 | 0.0550 | **0.0380** | 0.0590 | **0.1281** |
| mlLGPR+AB+RE+PC | 0.0966 | 0.0732 | 0.1394 | 0.0677 | 0.0515 | 0.0625 | 0.1793 |
| mlLGPR+AB+PE+PC | 0.1029 | 0.0899 | 0.1441 | 0.0914 | 0.1148 | 0.0903 | 0.1820 |
| mlLGPR+AB+RE+PE+PP | 0.2019 | 0.2070 | 0.2142 | 0.1876 | 0.1884 | 0.1880 | 0.2299 |
| mlLGPR+AB+RE+PE+PP | 0.2894 | 0.2993 | 0.2953 | 0.2736 | 0.2530 | 0.2755 | 0.2838 |
| mlLGPR+AB+RE+PE+PC | 0.0954 | 0.0816 | 0.1441 | 0.0673 | 0.0451 | 0.0641 | 0.1806 |
| mlLGPR+AB+RE+PE+PP+PC | 0.2003 | 0.2063 | 0.2209 | 0.1924 | 0.1924 | 0.1928 | 0.2317 |
| Methods | Average Precision Score ↑ | | | | | | |
| | EcoCyc | HumanCyc | AraCyc | YeastCyc | LeishCyc | TrypanoCyc | SixDB |
| mlLGPR+AB | 0.5478 | 0.5610 | 0.7390 | 0.5000 | 0.2316 | 0.3873 | 0.7323 |
| mlLGPR+AB+RE | **0.6205** | 0.6373 | 0.7275 | **0.6410** | 0.4293 | **0.5414** | 0.7412 |
| mlLGPR+AB+PP | 0.2755 | 0.2508 | 0.3926 | 0.2303 | 0.1037 | 0.1855 | 0.4300 |
| mlLGPR+AB+PE | 0.5473 | 0.5773 | 0.7495 | 0.5048 | 0.2257 | 0.3843 | 0.7402 |
| mlLGPR+AB+PC | 0.5618 | 0.5673 | 0.7810 | 0.5113 | 0.2265 | 0.4217 | 0.7650 |
| mlLGPR+AB+RE+PP | 0.2795 | 0.2536 | 0.3845 | 0.2375 | 0.1081 | 0.1885 | 0.4322 |
| mlLGPR+AB+RE+PE | 0.6187 | 0.6686 | 0.7372 | 0.6480 | **0.4731** | 0.5455 | 0.7561 |
| mlLGPR+AB+RE+PC | 0.6019 | **0.6926** | **0.7992** | 0.6330 | 0.3862 | 0.5362 | **0.7761** |
| mlLGPR+AB+PE+PC | 0.5681 | 0.5844 | 0.7645 | 0.4969 | 0.2188 | 0.4223 | 0.7727 |
| mlLGPR+AB+RE+PE+PP | 0.3241 | 0.3000 | 0.4730 | 0.2761 | 0.1309 | 0.2283 | 0.5122 |
| mlLGPR+AB+RE+PE+PP | 0.2706 | 0.2482 | 0.3870 | 0.2301 | 0.1068 | 0.1873 | 0.4309 |
| mlLGPR+AB+RE+PE+PC | 0.6065 | 0.6466 | 0.7744 | 0.6277 | 0.4237 | 0.5291 | 0.7715 |
| mlLGPR+AB+RE+PE+PP+PC | 0.3299 | 0.2997 | 0.4580 | 0.2701 | 0.1285 | 0.2244 | 0.5084 |
| Methods | Average Recall Score ↑ | | | | | | |
| | EcoCyc | HumanCyc | AraCyc | YeastCyc | LeishCyc | TrypanoCyc | SixDB |
| mlLGPR+AB | **0.9511** | 0.9068 | 0.7608 | **0.9258** | 0.9770 | 0.9429 | 0.6775 |
| mlLGPR+AB+RE | 0.9055 | 0.8566 | 0.7275 | 0.8734 | 0.9080 | 0.8971 | 0.6774 |
| mlLGPR+AB+PP | 0.8176 | 0.8280 | **0.7961** | 0.8559 | 0.8391 | 0.8800 | 0.7696 |
| mlLGPR+AB+PE | 0.9414 | **0.9104** | 0.7569 | 0.9170 | **0.9885** | **0.9486** | 0.6795 |
| mlLGPR+AB+PC | 0.6515 | 0.6344 | 0.4196 | 0.6900 | 0.8851 | 0.8000 | 0.3827 |
| mlLGPR+AB+RE+PP | 0.8339 | 0.8280 | 0.7765 | 0.8690 | 0.8736 | 0.9029 | **0.7768** |
| mlLGPR+AB+RE+PE | 0.8827 | 0.8459 | 0.7314 | 0.8603 | 0.9080 | 0.8914 | 0.6904 |
| mlLGPR+AB+RE+PC | 0.6059 | 0.6057 | 0.4137 | 0.6026 | 0.8391 | 0.7200 | 0.3820 |
| mlLGPR+AB+PE+PC | 0.6384 | 0.6452 | 0.4137 | 0.6900 | 0.9080 | 0.8229 | 0.3923 |
| mlLGPR+AB+PP+PC | 0.6091 | 0.6559 | 0.5333 | 0.6594 | 0.7931 | 0.7200 | 0.5053 |
| mlLGPR+AB+RE+PE+PP | 0.8143 | 0.8423 | 0.7922 | 0.8603 | 0.8621 | 0.8914 | 0.7758 |
| mlLGPR+AB+RE+PE+PC | 0.6124 | 0.5771 | 0.4039 | 0.6332 | 0.8621 | 0.6743 | 0.3776 |
| mlLGPR+AB+RE+PE+PP+PC | 0.6287 | 0.6487 | 0.5137 | 0.6594 | 0.7931 | 0.7257 | 0.5074 |
| Methods | Average F1 Score ↑ | | | | | | |
| | EcoCyc | HumanCyc | AraCyc | YeastCyc | LeishCyc | TrypanoCyc | SixDB |

*(Continued)*

**Table 3.** (Continued)

| | | | | | | | |
|---|---|---|---|---|---|---|---|
| mlLGPR+AB | 0.6952 | 0.6932 | 0.7498 | 0.6493 | 0.3744 | 0.5491 | 0.6754 |
| mlLGPR+AB+RE | **0.7364** | 0.7309 | 0.7275 | **0.7394** | 0.5830 | 0.6753 | 0.6938 |
| mlLGPR+AB+PP | 0.4122 | 0.3850 | 0.5259 | 0.3630 | 0.1846 | 0.3065 | 0.5386 |
| mlLGPR+AB+PE | 0.6922 | 0.7065 | **0.7532** | 0.6512 | 0.3675 | 0.5470 | 0.6802 |
| mlLGPR+AB+PC | 0.6033 | 0.5990 | 0.5459 | 0.5874 | 0.3607 | 0.5523 | 0.4683 |
| mlLGPR+AB+RE+PP | 0.4186 | 0.3882 | 0.5143 | 0.3730 | 0.1924 | 0.3119 | 0.5422 |
| mlLGPR+AB+RE+PE | 0.7275 | **0.7468** | 0.7343 | 0.7392 | **0.6220** | **0.6768** | **0.7098** |
| mlLGPR+AB+RE+PC | 0.6039 | 0.6463 | 0.5452 | 0.6174 | 0.5290 | 0.6146 | 0.4853 |
| mlLGPR+AB+PE+PC | 0.6012 | 0.6133 | 0.5369 | 0.5777 | 0.3527 | 0.5581 | 0.4779 |
| mlLGPR+AB+PP+PC | 0.4231 | 0.4117 | 0.5014 | 0.3892 | 0.2248 | 0.3466 | 0.4857 |
| mlLGPR+AB+RE+PE+PP | 0.4062 | 0.3834 | 0.5199 | 0.3631 | 0.1901 | 0.3095 | 0.5407 |
| mlLGPR+AB+RE+PE+PC | 0.6094 | 0.6098 | 0.5309 | 0.6304 | 0.5682 | 0.5930 | 0.4805 |
| mlLGPR+AB+RE+PE+PP+PC | 0.4327 | 0.4100 | 0.4843 | 0.3832 | 0.2212 | 0.3428 | 0.4847 |

## Robustness

**Experimental setup.** *Robustness* also known as *accuracy loss rate* was determined for mlLGPR-EN with AB, RE and PE feature sets using the intact Synset-1 dataset and a "corrupted" or noisy version of the Synset-2 dataset. Relative Loss of Accuracy (RLA) and equalized loss of accuracy (ELA) scores [42] were used to describe the expected behavior of mlLGPR-EN in relation to introduced noise. The ELA score explained in Section 2 in S3 Appendix, encompasses i)- the robustness of a model determined at a controlled noise threshold $\rho$, and ii)- the performance of a model without noise, i.e., $s(M_0)$, where $s$ represents the F1 score for a model $M_0$ without noise (any performance metrics can be employed). A low robustness score indicates that model continues to exhibit good performance with increasing background noise.

**Experimental results.** Table 4 indicates robustness test scores. mlLGPR-EN with introduced noise performed better for HumanCyc (−0.0502), YeastCyc (−0.0301), LeishCyc (−0.1189), and TrypanoCyc (−0.0151), but was less robust for AraCyc (0.0416) and SixDB (0.0470) based on $RLA_\rho$ scores. This suggests that noise inversely correlates with the pathway size. The more pathways present within a dataset can upset correlations among features. However, the impact of negative correlations is minimized when a dataset contains fewer pathways. Note that the average number of ECs associated with pathways has little or negligible effects on robustness.

**Table 4. Performance and robustness scores for mlLGPR-EN with AB, RE and PE feature sets trained on both Synset-1 and Synset-2 training sets at 0 and $\rho$ noise.** The best performance scores are highlighted in bold. The '↓' indicates the lower score is better while '↑' indicates the higher score is better.

| Dataset | Average F1 Score ↑ | | Robustness Score ↓ | | |
|---|---|---|---|---|---|
| | $mlLGPR\text{-}EN_0$ | $mlLGPR\text{-}EN_\rho$ | $RLA_\rho$ | $s(M_0)$ | $ELA_\rho$ |
| EcoCyc | **0.7280** | 0.7275 | 0.0007 | 0.3736 | 0.3743 |
| HumanCyc | 0.7111 | **0.7468** | −0.0502 | 0.4063 | 0.3561 |
| AraCyc | **0.7662** | 0.7343 | 0.0416 | 0.3051 | 0.3468 |
| YeastCyc | 0.7176 | **0.7392** | −0.0301 | 0.3935 | 0.3634 |
| LeishCyc | 0.5559 | **0.6220** | −0.1189 | 0.7989 | 0.6800 |
| TrypanoCyc | 0.6667 | **0.6768** | −0.0151 | 0.4999 | 0.4848 |
| SixDB | **0.7448** | 0.7098 | 0.0470 | 0.3426 | 0.3896 |

Taken together, the RLA and ELA results for T1 golden datasets indicate that mlLGPR-EN trained on noisy datasets is robust to perturbation. This is a prerequisite for developing supervised ML methods tuned for community-level pathway prediction.

## Pathway prediction potential

**Experimental setup.** Pathway prediction potential of mlLGPR-EN with AB, RE and PE feature sets trained on Synset-2 training set was compared to four additional prediction methods including Baseline, Naïve v1.2 [17], MinPath v1.2 [17] and PathoLogic v21 [10] on T1 golden datasets using the settings and metrics described above. For community-level pathway prediction on the T4 datasets including symbiont, CAMI low complexity, and HOTS datasets, mlLGPR-EN and PathoLogic (without taxonomic pruning) results were compared.

**Experimental results.** Table 5 shows performance scores for each pathway prediction method tested. The BASELINE, Naïve, and MinPath methods infer many false positive pathways across the T1 golden datasets, indicated by high recall with low precision and F1 scores. In contrast, high precision and F1 scores were observed for PathoLogic and mlLGPR-EN across the T1 golden datasets. Although both methods gave similar results, PathoLogic F1 scores for EcoCyc (0.7631), YeastCyc (0.7890) and SixDB (0.7479) exceeded those for

**Table 5. Pathway prediction performance between methods using T1 golden datasets.** mlLGPR-EN: the mlLGPR with elastic net penalty, L2: AB: abundance features, RE: reaction evidence features, and PE: pathway evidence features. For each performance metric, '↓' indicates the lower score is better while '↑' indicates the higher score is better.

| Methods | Hamming Loss ↓ | | | | | | |
|---|---|---|---|---|---|---|---|
| | EcoCyc | HumanCyc | AraCyc | YeastCyc | LeishCyc | TrypanoCyc | SixDB |
| BASELINE | 0.2217 | 0.2486 | 0.3230 | 0.2458 | 0.1591 | 0.2526 | 0.3096 |
| Naïve | 0.3856 | 0.4113 | 0.4592 | 0.4216 | 0.3215 | 0.4319 | 0.4392 |
| MinPath | 0.2257 | 0.2530 | 0.3266 | 0.2482 | 0.1615 | 0.2561 | 0.3124 |
| PathoLogic | **0.0610** | **0.0633** | 0.1188 | **0.0424** | **0.0368** | **0.0424** | **0.1141** |
| mlLGPR-EN (+AB+RE+PE) | 0.0804 | **0.0633** | **0.1069** | 0.0550 | 0.0380 | 0.0590 | 0.1281 |
| Methods | Average Precision Score ↑ | | | | | | |
| | EcoCyc | HumanCyc | AraCyc | YeastCyc | LeishCyc | TrypanoCyc | SixDB |
| BASELINE | 0.3531 | 0.3042 | 0.3832 | 0.2694 | 0.1779 | 0.2153 | 0.4145 |
| Naïve | 0.2384 | 0.2081 | 0.3035 | 0.1770 | 0.0968 | 0.1382 | 0.3357 |
| MinPath | 0.3490 | 0.3004 | 0.3806 | 0.2675 | 0.1758 | 0.2129 | 0.4124 |
| PathoLogic | **0.7230** | **0.6695** | 0.7011 | **0.7194** | **0.4803** | **0.5480** | 0.7522 |
| mlLGPR-EN (+AB+RE+PE) | 0.6187 | 0.6686 | **0.7372** | 0.6480 | 0.4731 | 0.5455 | **0.7561** |
| Methods | Average Recall Score ↑ | | | | | | |
| | EcoCyc | HumanCyc | AraCyc | YeastCyc | LeishCyc | TrypanoCyc | SixDB |
| BASELINE | **0.9902** | **0.9713** | **0.9843** | **1.0000** | **1.0000** | **1.0000** | **0.9860** |
| Naïve | **0.9902** | **0.9713** | **0.9843** | **1.0000** | **1.0000** | **1.0000** | **0.9860** |
| MinPath | **0.9902** | **0.9713** | **0.9843** | **1.0000** | **1.0000** | **1.0000** | **0.9860** |
| PathoLogic | 0.8078 | 0.8423 | 0.7176 | 0.8734 | 0.8391 | 0.7829 | 0.7499 |
| mlLGPR-EN (+AB+RE+PE) | 0.8827 | 0.8459 | 0.7314 | 0.8603 | 0.9080 | 0.8914 | 0.6904 |
| Methods | Average F1 Score ↑ | | | | | | |
| | EcoCyc | HumanCyc | AraCyc | YeastCyc | LeishCyc | TrypanoCyc | SixDB |
| BASELINE | 0.5205 | 0.4632 | 0.5516 | 0.4245 | 0.3021 | 0.3543 | 0.5784 |
| Naïve | 0.3843 | 0.3428 | 0.4640 | 0.3007 | 0.1765 | 0.2429 | 0.4939 |
| MinPath | 0.5161 | 0.4589 | 0.5489 | 0.4221 | 0.2990 | 0.3511 | 0.5763 |
| PathoLogic | **0.7631** | 0.7460 | 0.7093 | **0.7890** | 0.6109 | 0.6447 | **0.7479** |
| mlLGPR-EN (+AB+RE+PE) | 0.7275 | **0.7468** | **0.7343** | 0.7392 | **0.6220** | **0.6768** | 0.7098 |

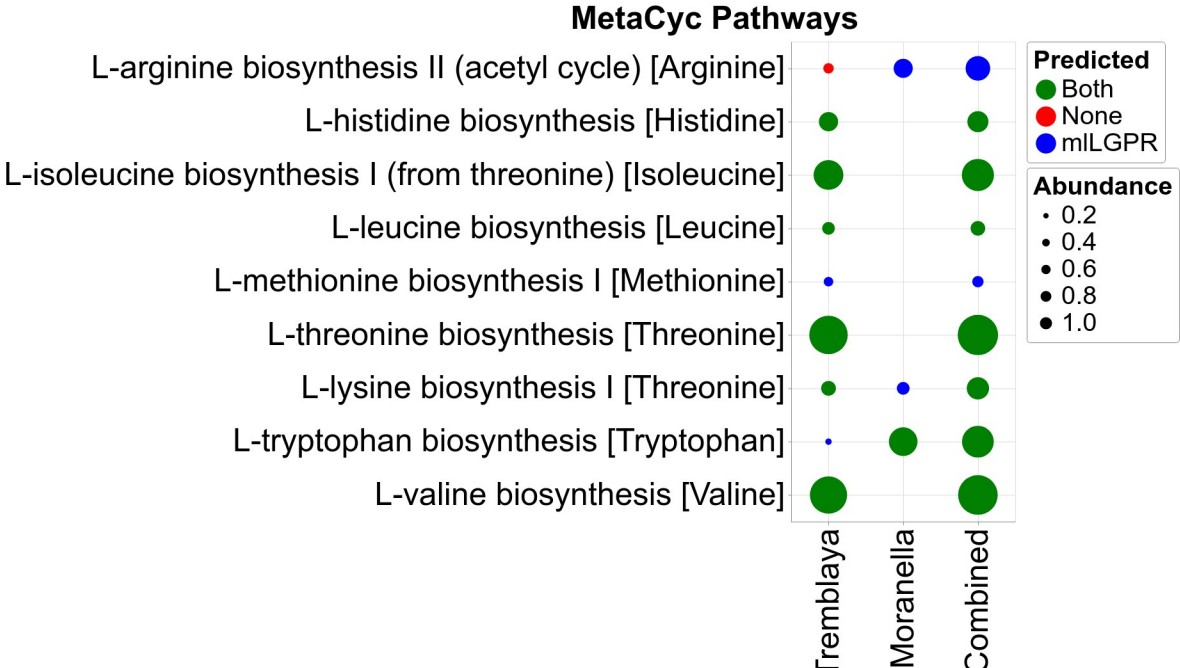

**Fig 5. Predicted pathways for symbiont datasets between mlLGPR-EN with AB, RE and PE feature sets and PathoLogic.** Red circles indicate that neither method predicted a specific pathway while green circles indicate that both methods predicted a specific pathway. Blue circles indicate pathways predicted solely by mlLGPR. The size of circles scales with reaction abundance information.

mlLGPR-EN. Conversely, mlLGPR-EN F1 scores for HumanCyc (0.7468), AraCyc (0.7343), LeishCyc (0.6220) and TrypanoCyc (0.6768) exceeded those for PathoLogic.

To evaluate mlLGPR-EN performance on distributed metabolic pathway prediction between two or more interacting organismal genomes a symbiotic system consisting of the reduced genomes for *Candidatus Moranella endobia* and *Candidatus Tremblaya princeps*, encoding a previously identified set of distributed amino acid biosynthetic pathways [24], was selected. mlLGPR-EN and PathoLogic were used to predict pathways on individual symbiont genomes and a composite genome consisting of both, and resulting amino acid biosynthetic pathway distributions were determined (Fig 5). mlLGPR-EN predicted 8 out of 9 expected amino acid biosynthetic pathways while PathoLogic recovered 6 on the composite genome. The missing pathway for phenylalanine biosynthesis (*L-phenylalanine biosynthesis I* was excluded from analysis because the associated genes were reported to be missing during the ORF prediction process. False positives were predicted for individual symbiont genomes in *Moranella* and *Tremblaya* using both methods although pathway coverage was low compared to the composite genome. Additional feature information restricting the taxonomic range of certain pathways or more restrictive pathway coverage could reduce false discovery on individual organismal genomes.

To evaluate pathway prediction performance of mlLGPR-EN on more complex community-level genomes the CAMI low complexity and HOTS datasets were selected. Table G in S3 Appendix shows performance scores for mlLGPR-EN on the CAMI dataset. Although recall was high (0.7827) precision and F1 scores were low when compared to the T1 golden datasets. Similar results were obtained for the HOTS dataset. In both cases it is difficult to validate most pathway prediction results without individual organismal genomes that can be replicated in culture. Moreover, the total number of expected pathways per dataset is relatively large,

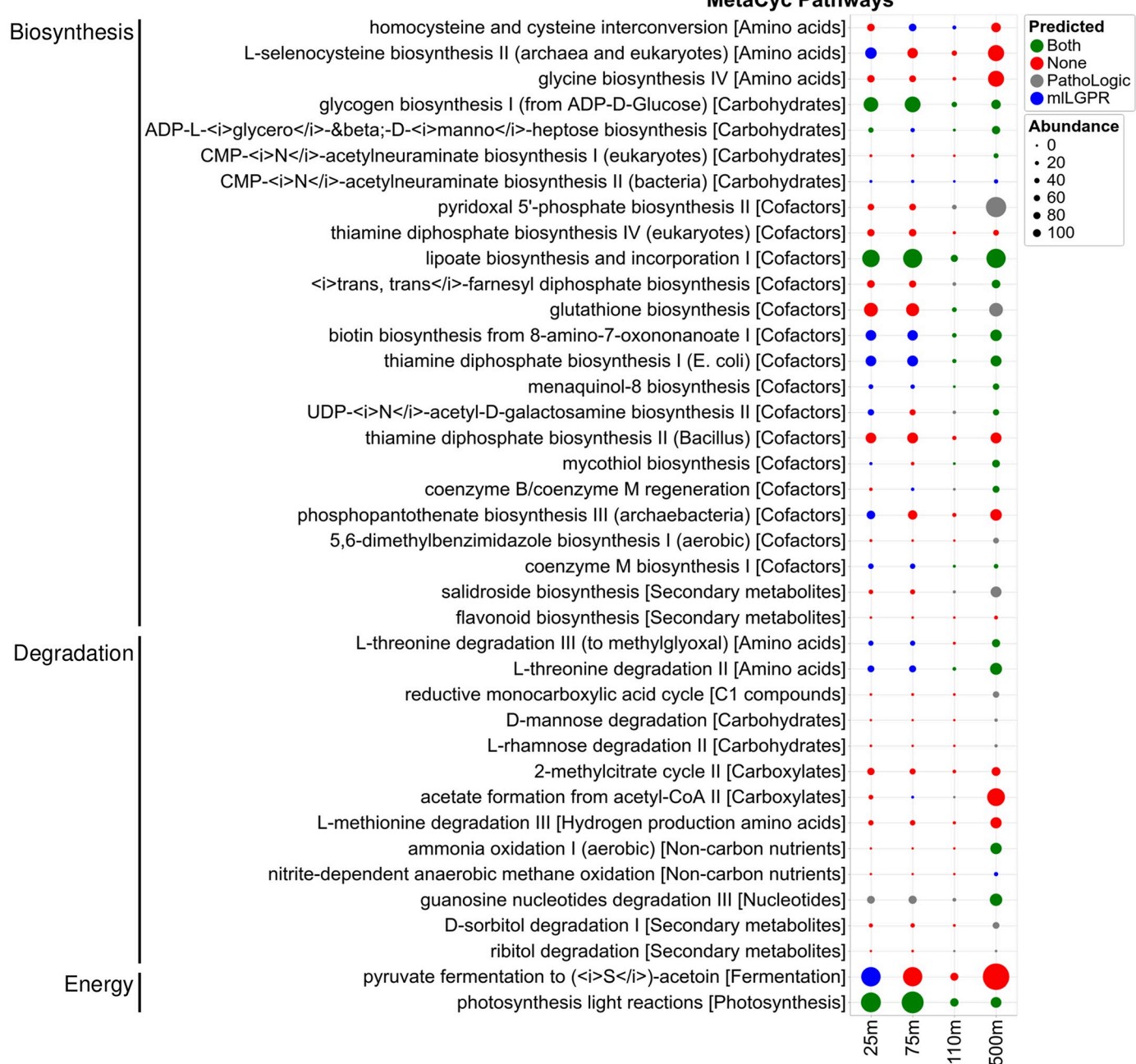

**Fig 6. Comparison of predicted pathways for HOTS datasets between mlLGPR-EN with AB, RE and PE feature sets and PathoLogic.** Red circles indicate that neither method predicted a specific pathway while green circles indicate that both methods predicted a specific pathway. Blue circles indicate pathways predicted solely by mlLGPR and gray circles indicate pathways solely predicted by PathoLogic. The size of circles scales with reaction abundance information.

encompassing metabolic interactions at different levels of biological organization. On the one hand, these open conditions confound interpretation of performance metrics while on the other they present numerous opportunities for hypothesis generation and testing. To better constrain this tension, mlLGPR-EN and PathoLogic prediction results were compared for a subset of 45 pathways previously reported in the HOTS dataset [14]. Fig 6 shows pathway distributions spanning sunlit and dark ocean waters predicted by PathoLogic and mlLGPR-EN,

grouped according to higher order functions within the MetaCyc classification hierarchy. Between 25 and 500 m depth intervals, 7 pathways were exclusively predicted by PathoLogic and 6 were exclusively predicted by mlLGPR-EN. Another 20 pathways were predicted by both methods, while 6 pathways were not predicted by either method including *glycine biosynthesis IV*, *thiamine diphosphate biosynthesis II and IV*, *flavanoid biosynthesis*, *2-methylcitrate cycle II* and *L-methionine degradation III*. In several instances, the depth distributions of predicted pathways were also different from those described in [14] including *L-selenocysteine biosythesis II* and *acetate formation from acetyl-CoA II*. It remains uncertain why current implementation of PathoLogic resulted in inconsistent pathway prediction results, although changes have accrued in PathoLogic rules and the structure of the MetaCyc classification hierarchy in the intervening time interval.

Taken together, the comparative pathway prediction results indicate that mlLGPR-EN performance equals or exceeds other methods including PathoLogic on organismal genomes but diminishes with dataset complexity.

## Discussion

We have developed mlLGPR, a new method using multi-label classification and logistic regression to predict metabolic pathways at different levels in the genomic information hierarchy (Fig 1). mlLGPR effectively maps annotated enzymatic reactions using EC numbers onto reference metabolic pathways sourced from the MetaCyc database. We provide a detailed open source process from features engineering and the construction of synthetic samples, on which the mlLGPR is trained, to performance testing on increasingly complex real world datasets including organismal genomes, nested symbionts, CAMI low complexity and HOTS. With respect to features engineering, five feature sets were re-designed from Dale and colleagues [18] to guide the learning process. Feature ablation studies demonstrated the usefulness of aggregating different combinations of feature sets using the elastic-net (EN) regularizer to improve mlLGPR prediction performance on golden datasets. Using this process we determined that abundance (AB), enzymatic reaction evidence (RE) and pathway evidence (PE) feature sets contribute disproportionately to mlLGPR-EN performance. After tuning several hyper-parameters to further improve mlLGPR performance, pathway prediction outcomes were compared to other methods including MinPath and PathoLogic. The results indicated that while mlLGPR-EN performance equaled or exceeded other methods including PathoLogic on organismal genomes, its performed more marginally on complex datasets. This is likely due to multiple factors including the limited validation information for community-level metabolism as well as the need for more subtle features engineering and algorithmic improvements.

Several issues were identified during testing and implementation that need to be resolved for improved pathway prediction outcomes using machine learning methods. While rich feature information is integral to mlLGPR performance, the current definition of feature sets relies on manual curation based on prior knowledge. We observed that in some instances the features engineering process is susceptible to noise resulting in low performance scores. Moreover, individual enzymes may participate in multiple pathways, e.g. multiple mapping problem, resulting in increased false discovery without additional feature sets that relate the presence and abundance of EC numbers to other factors. This problem has been partially addressed by designing features based on side knowledge of a pathway, such as information about "key-reactions" in pathways that increase the likelihood that a given pathway is present. Additional factors including taxonomy, gene expression, or environmental context should also be considered in features engineering for specific information structures. For example,

taxonomic constraints on metabolic potential are difficult to use when predicting pathways at the community level given the limited number of closed genomes present in the data. In contrast, environmental context information such as physical and chemical parameter data could be used to constrain specific metabolic potential e.g. aerobic versus anaerobic or light- versus dark-dependent processes. Missing EC numbers also present a challenge especially when trying to define "key-reactions" in pathways with less biochemical validation. An alternative method might be to apply *representational learning* [43], e.g. learning features from data automatically that can be supplemented with side knowledge to improve pathway prediction outcomes. Finally, alternative algorithms used to analyze high dimensional datasets such as graph based learning [44] has potential to provide even more accurate models needed to inform future experimental design and pathway engineering efforts.

## Supporting information

**S1 Appendix. Mathematical derivations of mlLGPR.** This file describes the process of deriving the objective cost function in Eq 9.
(PDF)

**S2 Appendix. Features used for mlLGPR.** This file describes features engineering aspects of the work. Given a set of enzymatic reactions with abundance information, we extract sets of features to capture salient aspects of metabolism for pathway inference.
(PDF)

**S3 Appendix. Additional experiments.** This file contains additional test results that are not presented in the main article including more in-depth information related to datasets and the ELA robustness metric.
(PDF)

**S1 Table. Pathway abundance information from symbiont data.** MetaCyc Pathway ID: The unique identifier for the pathway as provided by MetaCyc; MetaCyc Pathway Name: The name of the pathway as outlined by MetaCyc; Moranella: the Moranella endosymbiont (GenBank NC-015735); Tremblaya: the Tremblaya endosymbiont (GenBank NC-015736); Composite: a composite genome consisting of both endosymbiont genomes. Each numeric value encodes the coverage information of a pathway associated with each endosymbiont or composite genome. The coverage is computed based on mapping enzymes onto true representations of each pathway and is within the range of [0, 1], where 1 indicates that all enzymes catalyzing reactions in a given pathway were identified while 0 means no enzymes were observed for a given pathway.
(TSV)

**S2 Table. Pathway abundance information from HOTS data.** MetaCyc Pathway ID: The unique identifier for the pathway as provided by MetaCyc; MetaCyc Pathway Name: The name of the pathway as outlined by MetaCyc; 25m: the 25 m depth interval in the HOTS water column; 75m: the 75m depth interval in the HOTS water column; 110m: the 110 m depth interval in the HOTS water column; and 500m: the 500 m depth interval in the HOTS water column. Each numeric value encodes abundance information for a given pathway associated with each depth interval. The abundance is expected pathway copies normalized based on mapping identified enzymes onto true representations of each selected pathway.
(TSV)

## Acknowledgments

We would like to thank Connor Morgan-Lang, Julia Glinos, Kishori Konwar and Aria Hahn for lucid discussions on the function of the mlLGPR model and all members of the Hallam Lab for helpful comments along the way.

## Author Contributions

**Conceptualization:** Abdur Rahman M. A. Basher, Steven J. Hallam.

**Formal analysis:** Abdur Rahman M. A. Basher, Ryan J. McLaughlin.

**Funding acquisition:** Steven J. Hallam.

**Investigation:** Abdur Rahman M. A. Basher, Ryan J. McLaughlin.

**Methodology:** Steven J. Hallam.

**Project administration:** Steven J. Hallam.

**Software:** Abdur Rahman M. A. Basher.

**Supervision:** Steven J. Hallam.

**Validation:** Abdur Rahman M. A. Basher, Ryan J. McLaughlin.

**Visualization:** Abdur Rahman M. A. Basher, Ryan J. McLaughlin, Steven J. Hallam.

**Writing – original draft:** Abdur Rahman M. A. Basher, Steven J. Hallam.

**Writing – review & editing:** Ryan J. McLaughlin, Steven J. Hallam.

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
