## [Decision Letter · Decision Letter 0]

11 May 2020

Dear Hallam,

Thank you very much for submitting your manuscript "Metabolic pathway inference using multi-label classification with rich pathway features" for consideration at PLOS Computational Biology.

As with all papers reviewed by the journal, your manuscript was reviewed by members of the editorial board and by several independent reviewers. In light of the reviews (below this email), we would like to invite the resubmission of a significantly-revised version that takes into account the reviewers' comments.

We cannot make any decision about publication until we have seen the revised manuscript and your response to the reviewers' comments. Your revised manuscript is also likely to be sent to reviewers for further evaluation.

Sincerely,

William Cannon

Guest Editor

PLOS Computational Biology

Mark Alber

Deputy Editor

PLOS Computational Biology

Reviewer's Responses to Questions

**Comments to the Authors:**

Reviewer #1: review is uploaded as an attachment

Reviewer #2: In this paper, Basher, A. et al. proposed a logistic regression method to infer metabolic pathways. Listed below are specific points of concerns that need to be addressed.

1. In line 113, the authors wrote that they considered five types of feature vectors based on the work of Dale et al. However, they did not describe the details about whether they used exactly the same features or expanded the features relative to the previous work. The features need to be compared in more detail. Since the previous work also used logistic regression, the authors are strongly encourage to explain what improvements have been made compared to the previous work.

2. It seems that the training data is important in learning. The authors should describe in more detail how the training data Synset-1 and Synset-2 have been prepared. How many organisms were used for each training data?

3. It is less clear what the authors want to show in Table 3 by evaluating with or without AB, RE, or PP? As far as this reviewer can see, the performance seems quite comparable regardless of the choice of the feature categories, such as AB, RE, or PP. Instead of using (or not using) the entire category of features, it might be more interesting to show what specific features are important for the performance. This reviewer strongly suggests that the authors evaluate the effects of more specific features, for example, some specific features in AB categories.

4. It seems that the authors did not compare their performance with Dale et al.’s work [18]. Dale et al. already used diverse machine learning methods to infer metabolic pathways. It seems that the predictor used by Dale et al. is considerably good. Please show how much improvement in accuracy was made by this work compared with Dale et al.’s classifiers.

5. Table 1 should be placed in landscape orientation. The domain information is hard to read.

6. What is the baseline method in Table 5?

**Have all data underlying the figures and results presented in the manuscript been provided?**

Reviewer #1: Yes

Reviewer #2: No: It needs more details about the training data, as described in comments.

PLOS authors have the option to publish the peer review history of their article (what does this mean?). If published, this will include your full peer review and any attached files.

Reviewer #1: No

Reviewer #2: No
---

## [Decision Letter · Decision Letter 1]

21 Jul 2020

Dear Hallam,

We are pleased to inform you that your manuscript 'Metabolic pathway inference using multi-label classification with rich pathway features' has been provisionally accepted for publication in PLOS Computational Biology.

Best regards,

William Cannon

Guest Editor

PLOS Computational Biology

Mark Alber

Deputy Editor

PLOS Computational Biology

Reviewer's Responses to Questions

**Comments to the Authors:**

Reviewer #1: I have now revisited the manuscript by Basher et al. The manuscript is much improved and I am satisfied with the changes made by the authors and their responses to my comments. The github page/readme is very much improved and will help users enormously. While I think there is still further scope in streamlining and improving the readme, I commend the authors for the improvements and think that further notes and changes can happen in due course as issues are raised by users. It is my opinion that this software has the potential to be widely used in the field.

I was also able to test the software and below are my runtime notes. I was able to understand the arguments and what they mean, and installation was straightforward.

Testing the program mlLGRP: https://github.com/hallamlab/mlLGPR

Accessed July 15, 2020. Tested on the Lab’s linux server.

PROCESSING

Example:

1 : no – see below

2: yes

3: yes

4: no – see below

5: yes

TRAINING:

Example:

1: works

2: works

3: with L2 regulation but I can’t find this file: mlLGPR_l2_ab_re_pe_pp_pc.pk

With L1 either there is not a new file being created, but it modifies the file named mlLGPR_en_ab_re_pe.pkl in mdpath from TrainingExample1

4: works

PREDICTING

Example:

1: This works. Suggestion in the phrasing. When I read "Make sure you obtain "mlLGPR_en_ab_re_pe.lists" and "mlLGPR_en_ab_re_pe.details" files corresponding a list of infered pathways and predicted pathways with score and abundance information. “ I thought I had to obtain those files first but what was meant is that this creates those files.

2: works

3: works

Overall, all the examples that are given work, and the instructions are a lot easier to understand than the previous time. I like that the Zenodo link has all the necessary files to make the program run. The processing part with the flatfiles was harder to understand and I wasn’t able to make ex. 1 and 4 of the Processing section work, but it wasn’t too bad since I didn’t have the PGDB but that’s ok because that’s what previously we needed a subscription too, and the others have provided the generated files “object.pkl, pathwayfeature.pkl, ecfeature.pkl, pathway_ec.pkl, reaction_ec.pkl, and ec2pathway.txt” in the Zenodo file. I didn’t necessarily understand all the files and how to interpret them. Otherwise, the doc on the github was sufficient to understand the arguments and what they mean.

Minor changes

Line 218: where -> were

Reviewer #2: The authors have more or less addressed all my concerns.

**Have all data underlying the figures and results presented in the manuscript been provided?**

Reviewer #1: Yes

Reviewer #2: Yes

PLOS authors have the option to publish the peer review history of their article (what does this mean?). If published, this will include your full peer review and any attached files.

Reviewer #1: No

Reviewer #2: No

---

## [Editor Report · Acceptance letter]

28 Aug 2020

PCOMPBIOL-D-20-00171R1 

Metabolic pathway inference using multi-label classification with rich pathway features

Dear Dr Hallam,

I am pleased to inform you that your manuscript has been formally accepted for publication in PLOS Computational Biology. Your manuscript is now with our production department and you will be notified of the publication date in due course.

With kind regards,

Laura Mallard
